# Sphingolipid and Endocannabinoid Profiles in Adult Attention Deficit Hyperactivity Disorder

**DOI:** 10.3390/biomedicines9091173

**Published:** 2021-09-06

**Authors:** Nathalie Brunkhorst-Kanaan, Sandra Trautmann, Yannick Schreiber, Dominique Thomas, Sarah Kittel-Schneider, Robert Gurke, Gerd Geisslinger, Andreas Reif, Irmgard Tegeder

**Affiliations:** 1Department of Psychiatry, Psychotherapy and Psychosomatic Medicine, Medical Faculty, Goethe-University Frankfurt, 60590 Frankfurt, Germany; nathalie.brunkhorst-kanaan@kgu.de (N.B.-K.); andreas.reif@kgu.de (A.R.); 2Institute of Clinical Pharmacology, Medical Faculty, Goethe-University Frankfurt, 60590 Frankfurt, Germany; trautmann@med.uni-frankfurt.de (S.T.); yannick.schreiber@itmp.fraunhofer.de (Y.S.); thomas@med.uni-frankfurt.de (D.T.); robert.gurke@itmp.fraunhofer.de (R.G.); geisslinger@em.uni-frankfurt.de (G.G.); 3Fraunhofer Institute for Translational Medicine and Pharmacology ITMP, 60596 Frankfurt, Germany; 4Department of Psychiatry, Psychosomatic Medicine and Psychotherapy, University Hospital Würzburg, 97080 Würzburg, Germany; kittel_s@ukw.de; 5Fraunhofer Cluster of Excellence for Immune Mediated Diseases (CIMD), 60596 Frankfurt, Germany

**Keywords:** attention deficit hyperactivity disorder, endocannabinoids, ceramides, bipolar disorder, major depression, tandem mass spectrometry

## Abstract

Genes encoding endocannabinoid and sphingolipid metabolism pathways were suggested to contribute to the genetic risk towards attention deficit hyperactivity disorder (ADHD). The present pilot study assessed plasma concentrations of candidate endocannabinoids, sphingolipids and ceramides in individuals with adult ADHD in comparison with healthy controls and patients with affective disorders. Targeted lipid analyses of 23 different lipid species were performed in 71 mental disorder patients and 98 healthy controls (HC). The patients were diagnosed with adult ADHD (n = 12), affective disorder (major depression, MD n = 16 or bipolar disorder, BD n = 6) or adult ADHD with comorbid affective disorders (n = 37). Canonical discriminant analysis and CHAID analyses were used to identify major components that predicted the diagnostic group. ADHD patients had increased plasma concentrations of sphingosine-1-phosphate (S1P d18:1) and sphinganine-1-phosphate (S1P d18:0). In addition, the endocannabinoids, anandamide (AEA) and arachidonoylglycerol were increased. MD/BD patients had increased long chain ceramides, most prominently Cer22:0, but low endocannabinoids in contrast to ADHD patients. Patients with ADHD and comorbid affective disorders displayed increased S1P d18:1 and increased Cer22:0, but the individual lipid levels were lower than in the non-comorbid disorders. Sphingolipid profiles differ between patients suffering from ADHD and affective disorders, with overlapping patterns in comorbid patients. The S1P d18:1 to Cer22:0 ratio may constitute a diagnostic or prognostic tool.

## 1. Introduction

Attention deficit hyperactivity disorder (ADHD) is characterized by persistent inappropriate levels of inattention, hyperactivity and impulsiveness [1]. It is a multifactorial trait in which genetic factors account for about 75% of the phenotypic variance, and environmental factors, including premature birth, perinatal hypoxia, maternal smoking or alcohol, were suggested to contribute to the remaining 25% [2,3]. ADHD may persist through adulthood in about 65% of the cases with varying severity [4]. Adult ADHD is frequently associated with several mental disorders, including major depression (MD) and bipolar disorder (BD) [5], and it is genetically correlated with mental disorders [6,7,8]. Like MD/BD, ADHD patients have an increased risk of obesity and metabolic diseases [9].

Recently, whole-genome sequencing studies identified genetic variations in genes involved in fatty acid metabolism in association with ADHD, including fatty acid desaturase *FADS3* and elongase *ELOVL5* [10] and fatty acid amide hydrolase (*FAAH*) [11]. FAAH catalyzes the breakdown of ethanolamide endocannabinoids, e.g., anandamide (AEA). The classical endocannabinoids, AEA, and arachidonoylglycerol (1/2-AG) are agonists of cannabinoid-1 (CB1) and CB2 receptors, but also activate “orphan” G-protein coupled receptors [12,13,14], such as G-protein coupled receptor, GPR55 and nuclear receptors such as peroxisome proliferator-activated receptors, PPARs [15]. CB1 is the predominant canonical cannabinoid receptor in the brain, but CB1 is also abundant in peripheral tissues such as the liver, muscle, and endothelium, whereas CB2 is mainly expressed by immune cells of myeloid origin. Endocannabinoids are highly versatile, and deregulation of CB1 and/or CB2 signaling have been implicated in pain, anxiety, depression, other mental diseases, and metabolic, cardiovascular, and liver diseases (among others) [16,17,18,19,20,21]. In addition, polymorphisms of CB1 (*Cnr1*) have been associated with ADHD [22], and ADHD patients report relief from cannabis [18,23], but it is unknown if the benefit arises from supplementation of a biological endocannabinoid deficiency [24]. Interestingly, a commentary in Neurology in 2009 hypothesized that endocannabinoid metabolism is altered in childhood ADHD [25].

In addition to fatty acid metabolism, ADHD genetics pointed to alteration of sphingolipid metabolism originating from variants of ceramide synthases 6 (*LASS6/Cers6*) and *LASS2/Cers2* [10], which generate C16 and C24 ceramides, respectively. The ceramide associations support the current view that sphingolipid allostasis contributes to the pathophysiology of mental disorders [26]. Plasma concentrations of long-chain ceramides are persistently increased in patients with MD or BD irrespective of their chain lengths (from C16 to C24) [27,28,29], increase with age and with the number of episodes, but not in direct association with the current episode [29]. The ceramide accumulation suggested a gain of function of ceramide synthases in MD/BD and revealed a link to metabolic diseases such as diabetes mellitus contributed by Cer6/LASS6 generated ceramides [30,31,32].

Ceramides are intermediates in the biosynthesis of complex sphingolipids such as sphingomyelins, cerebrosides, and gangliosides [33,34]. They are generated through three major pathways, de novo synthesis via ceramide synthases [35], via reacylation of sphingosine [36], and by hydrolysis of sphingomyelins by sphingomyelinases (SMases) [34]. There are also three major pathways for ceramide breakdown yielding hexosylceramides, ceramide-1-phosphate, or sphingosine. The latter is the precursor of sphingosine-1-phosphate (S1Pd18:1), activating at least five distinct G-protein coupled receptors resulting in complex immune-regulatory effects. The functional antagonist, fingolimod, is a disease-modifying drug for immune-mediated diseases of the nervous system. Maternal immune activation in animal models elicits behavioral and neural outcomes consistent with ADHD in the offspring [37], suggesting that neuroinflammation contributes to the risk factors for ADHD. S1P has not yet been studied in this context.

Based on our previous studies in MD/BD patients [29] and patients with neuroinflammatory [38] and neurodegenerative diseases [39], we hypothesized that a set of sphingolipids and fatty acids would demark lipid profiles of adult ADHD patients and possibly predict comorbid outcomes. We, therefore, analyzed plasma concentrations of 23 lipid species (analytes and abbreviations in Appendix A) in ADHD patients, MD/BD patients, and patients with ADHD-MD/BD comorbidities versus healthy controls.

## 2. Materials and Methods

### 2.1. Participants and Biomaterial Sampling

Patients with adult attention deficit hyperactivity disorder (n = 12 ADHD, n = 37 ADHD plus MD/BD, denoted as ADHD+) and patients with major depression (MD, n = 16) or bipolar disorder (BD) (n = 6) and healthy controls (n = 98) (Table 1, Table 2 and Table 3) were consecutively recruited from inpatients of the Department of Psychiatry, Psychotherapy and Psychosomatic Medicine (patients) and from staff members (psychiatry or pharmacology) of the University Hospital Frankfurt. The estimate of the number of patients and controls was based on a previous study. We have analyzed plasma lipid profiles in an independent cohort of MD/BD patients and healthy controls [29]. The protocol followed an observational parallel-group design. Patients’ inclusion criteria were age ≥ 18 years and a clinically verified diagnosis of ADHD, major depression, or bipolar disorder based on ICD10 criteria and validated by two specialists independently, and informed written consent to participate. Patients were treated according to guidelines for their respective disorder and episode with drugs approved for ADHD, antidepressants, antipsychotics and/or mood stabilizers or antiepileptic agents. S-ketamine and electroconvulsive therapy (ECT) was applied in every two patients. Five of the ADHD patients reportedly used cannabis occasionally. For controls, inclusion criteria were age ≥ 18 years, no current medical condition queried by medical interview, and no drug intake for at least one week except contraceptives, vitamins, or L-Thyroxin, and informed written consent. Acetylsalicylic acid (100 mg) and antihypertensive drugs were not exclusionary in controls ≥ 50 years. Human blood samples were collected in K^+^-EDTA tubes (Microvette Sarstedt, Germany), kept on ice, and centrifuged in a tabletop centrifuge (Eppendorf) at 3000 rcf and 4 °C for 10 min within 15 min of blood collection. Plasma aliquots were stored immediately at −80 °C until analysis. Informed written consent was obtained from all participants. Sampling and data analyses adhered to the Declaration of Helsinki and were approved by the Ethics Committee of the Medical Faculty of the Goethe University (Approval # 425/14, approval date 4 March 2015). Demographic data, comorbidities, and treatments are summarized in Table 1, Table 2 and Table 3. Individualized drug treatments are included in Appendix A.

### 2.2. Clinical Assessment and Questionnaires

The severity of depression was assessed with the Beck Depression Inventory (BDI), which is a 21-question multiple-choice self-report inventory for measuring the severity of depression. Patients with bipolar disorder were additionally queried with the Young Mania Rating Scale (YMRS) to assess the severity of manic symptoms. The psychometric assessment encompasses 11 major clinical symptoms characterizing the last 48 h [40]. The ADHD diagnosis was based on a diagnostic test battery including the DIVA 2.0, a semi-structured diagnostic interview [41,42], Wender–Reimherr Interview (WRI) [43], and a short form of the Wender–Utah Rating Scale (WURS-k) [44,45].

### 2.3. Analysis of Lipid Signaling Molecules

Bioactive lipids, including sphingolipids and ceramides, and endocannabinoids, were analyzed in plasma by liquid chromatography-electrospray ionization tandem mass spectrometry (LC-ESI-MS/MS) as described in recent previous publications [29,39,46].

Briefly, the determination of endocannabinoids, arachidonoyl ethanolamide (AEA), 2-arachidonoyl glycerol (2-AG), 1-arachidonoyl glycerol (1-AG), oleoyl ethanolamide (OEA), and palmitoylethanolamide (PEA) was performed as described in detail in Reference [46]. Analytes are listed in Appendix A. The extraction and separation were done by a liquid-liquid-extraction (LLE) protocol combined with ultra-high-performance liquid chromatography-tandem mass spectrometry (UHPLC-MS/MS). Plasma samples were thawed in a refrigerator and processed on ice and were extracted using a mixture of ethyl acetate/hexane (9:1, *v/v*) after adding the internal standard. The organic phase was removed and evaporated. The sample was reconstituted in acetonitrile and injected into the LC-MS/MS system. The LC-MS/MS system consisted of a triple quadrupole mass spectrometer QTRAP 6500+ (Sciex, Darmstadt, Germany) equipped with a Turbo Ion Spray source and an Agilent 1290 Infinity LC-system with binary HPLC pump, column oven, and autosampler (Agilent, Waldbronn, Germany). It operated in positive electrospray ionization mode. The chromatographic separation was performed on an Acquity UPLC BEH C18 2.1 × 100 mm column (particle size of 1.7 µm, Waters, Eschborn, Germany).

The determination of sphingoid bases and ceramides was performed as described in detail in Reference [47]. The analytes Cerd18:0/16:0, Cerd18:0/18:0, Cerd18:0/18:1, Cerd18:0/24:0, Cerd18:0/24:1, Cerd18:1/14:0, Cerd18:1/16:0, Cerd18:1/18:0, Cerd18:1/18:1, Cerd18:1/20:0, Cerd18:1/22:0, Cerd18:1/24:0, Cerd18:1/24:1, GlcCerd18:1/16:0, GlcCerd18:1/18:0, GlcCerd18:1/18:1, GlcCerd18:1/24:1, LacCerd18:1/16:0, LacCer d18:1/18:0, LacCerd18:1/18:1, LacCerd18:1/24:0, LacCerd18:1/24:1, SPHd18:0, SPHd18:1, SPH d20:0, SPHd20:1, S1Pd16:1, S1Pd18:0 and S1Pd18:1 (Appendix A) were determined in human plasma using LLE in combination with UHPLC-MS/MS. For the quantification, 10 µL of plasma was extracted using 200 µL extraction buffer (citric acid 30 mM, disodium hydrogen phosphate 40 mM, pH 4.2), 20 µL internal standard, and 600 µL of a mixture of methanol: chloroform: hydrochloric acid (15:83:2, *v/v/v*). The extract was divided into two aliquots, one for the determination of ceramides and one for the determination of sphingoid bases. The extracts were evaporated at 45 °C under a slight stream of nitrogen and reconstituted in 50 µL solvent. For ceramides, tetrahydrofuran/water (9:1, *v/v*) with 0.2% formic acid and 10 mM ammonium formate were used. For sphingoid bases, methanol containing 5% formic acid was used. The LC-MS/MS system consisted of a triple quadrupole mass spectrometer QTRAP 5500 (Sciex, Darmstadt, Germany) equipped with a Turbo Ion Spray source and Agilent 1290 Infinity LC-system as above. It was operated in positive electrospray ionization mode. The chromatographic separation of ceramides was done on a Zorbax RRHD Eclipse Plus C18 column (1.8 µm 50 × 2.1 mm ID), and sphingoid bases were separated on a Zorbax Eclipse Plus C8 UHPLC column (1.8 µm 30 × 2.1 mm ID; both Agilent, Waldbronn, Germany).

After measuring the calibration standards, quality control samples of three different concentrations (low, middle, high) were run as initial and final. The concentrations of calibration standards, quality controls, and samples were evaluated by Analyst software 1.7.1 and MultiQuant Software 3.0.3 (both Sciex, Darmstadt, Germany) using the internal standard method. Calibration curves were calculated by linear or quadratic regression with 1/x weighting or 1/x^2^ weighting. Acceptance criteria and quality assurance measures were as described in [46].

### 2.4. Statistical Analyses

Lipid concentrations are presented as scatter plots with mean ± standard deviation (SD) or box-scatter plots, where the box is the interquartile range, and the whiskers show minimum to maximum. Lipid profiles are presented as mean ± SD or sem as specified in the figure legends. Data were analyzed with SPSS 25, Origin Pro 2020, and GraphPad Prism 8.4 and 9.0. According to Schapiro Wilk and Anderson Darling, the data distributions were assessed according to goodness of fit criteria for normal distribution and log normal distribution. Lipids were log-normally distributed and therefore log2 transformed for further analyses. Partial least square analysis (PLS) was used to reduce dimensionality and identify the factors, which contributed most to the variance, and discriminated best between groups. Groups (HC, ADHD, ADHD+, MD/BD), gender, age class, and BMI class were considered as independent factors. ADHD+ refers to patients with a primary diagnosis of ADHD who suffered from a mood disorder as comorbidity. In addition, linear canonical discriminant analysis (CanDisc) was used to assess the predictability of group membership based on CanDisc scores. Chi-square automatic interaction detection (CHAID) was used to generate decision trees based on Bonferroni adjusted significance testing. Lipids were introduced as independent variables and gender as an influencing variable.

We used univariate or multivariate analyses of variance (ANOVA, MANOVA), or *t*-tests according to the data subgroup structure and distribution for between-group comparison. For multivariate analyses, the factors were “group” by “gender,” either with four groups (HC, ADHD, ADHD+, MD/BD) or three groups, where ADHD patients were summarized to gain power. For univariate analyses of “group”, age and body mass index were introduced as covariates. To address the younger ages of ADHD patients, a subset of young age-matched HC ≤ 40 years of age was compared with ADHD patients. For cluster analyses, PLS, and lipid profiles, lipid concentrations were normalized to the 75% quantile of the HC to allow for a combined analysis because the concentrations differ by several orders of magnitude.

Further analyses consisted of χ^2^ statistics, paired t-tests of candidate two lipids, and linear regression analyses to address age and BMI and interrelationships of two lipids. In case of significant results of ANOVAs, groups were mutually compared using *t*-tests, and *p*-values were adjusted according to the procedures of Dunnett (versus one control group) or Šidák. The alpha level was set at 0.05 for all comparisons, and asterisks in the figures refer to adjusted *p*-values.

## 3. Results

### 3.1. Canonical Discriminant Analysis Separate ADHD from MD/BD and Healthy Controls

We first used canonical discriminant analysis to assess whether patients suffering from ADHD could be separated from healthy controls and patients suffering from mood disorders (MD/BD) based on their lipid profiles. The CanDisc score plots (Figure 1A) show that ADHD patients differed from HC and MD/BD patients, irrespective of having or not having comorbidity. The first two CanDisc functions accounted for 67.3% and 22.2% of the variance. Function-1 mainly separated HC from patients, and function-2 separated ADHD from MD and BD. Group membership was correctly predicted for 91% of HC, 67% of ADHD, 60% of ADHD+ and 73% of MD/BD. The tree map of the CanDisc structure matrix for function-1 (Figure 1B) shows the relative importance of the different lipids for discriminating between HC and patients. The key candidates are Cer22:0 and S1P d18:1. The respective tree map for function-2 is presented in Appendix A. The groups’ lipid profiles depicted as polar and as area plots (Figure 1C,D) show that ADHD patients have increased S1Pd18:1, S1Pd18:0, and AEA, whereas MD/BD patients mainly have increased ceramides of different chain lengths. ADHD patients with MD/BD comorbidity ranged between the other two patient groups, i.e., they have the elevated S1P d18:1 plus elevated ceramides. Still, the levels were not as high as in the respective mono-diagnosis group.

Considering group differences, overall abundance and variability, the ceramide importance ranking was Cer22:0 > Cer24:1 > Cer20:0 > Cer18:0 > Cer16:0 > Cer24:0 > Cer14:0 (all d18:1 ceramides). CHAID analysis revealed that patients could be separated from HC based on Cer22:0 only (Figure 1E). Adding S1P d18:1 to the model to separate cases with high or mid Cer22:0 into further subgroups did not improve the predictability. For the analyses shown in Figure 1F, ADHD and ADHD+ were combined to gain power.

### 3.2. Influence of Age and Sex and Drug Treatments

All ceramides linearly increase with age, and Cer18:0 increases with BMI (Appendix A). Since ADHD patients without comorbidities were younger (≤40 years old), this could lead to spurious associations. To address this potential, confound, data were reanalyzed with a subgroup of HC ≤ 40 years of age. The result was robust for the key candidate lipids profiles. Ceramides of ADHD patients without comorbidities (ADHD-only) were in the normal range, and AEA and S1Pd18:1/S1Pd18:0 were elevated.

Further CHAID analyses revealed that males could be separated from females based on 1-AG. The significant separation disappeared when ADHD+ patients were excluded. Hence, the sex difference originated from ADHD+ males. Because of the impact of sex on endocannabinoids [48], statistics were run as MANOVA for the factors “group” X “gender” (Figure 2). For subsequent group-wise comparisons, males and females were combined. Ceramides were significantly increased in MD/BD patients without or with ADHD, and S1Pd18:1 and S1Pd18:0 were increased in ADHD. AEA was high in ADHD and 1-AG in ADHD+. However, AEA and AG are both agonists of classical cannabinoid receptors, with distinct site-specific functions in cannabinoid signaling circuits.

To assess putative influences of drug treatments, we compared patients receiving or not receiving drugs of the key drug classes, which were “ADHD-drugs” (MPH, lisdexamfetamine, atomoxetine, guanfacine), antidepressants (mostly venlafaxine, buproprione, TCA, or SSRI), and antipsychotics (typical and atypical neuroleptic agents) (Appendix A). A detailed map of individual drugs is presented in Appendix A. ADHD and ADHD+ (with comorbidity) were analyzed together. There was no significant influence of drugs on the key lipids AEA, Cer22:0, and S1P 18:1. It has to be considered that drugs and combinations of drugs reflect the course and severity of the disease, and owing to the diversity, sample sizes for a specific compound were low. Considering all ceramide species, ADHD+ and MD/BP patients receiving an antidepressant had higher levels of Cer18:0 than patients of these groups who did not receive an antidepressant (Appendix A). The result agrees with our previous study in MD/BP patients [29] and did not result from different BMI (ADHD+ or MD/BP without antidepressant BMI 26.176 ± 5.374; with antidepressant BMI 26.510 ± 6.330).

### 3.3. Switch of Lipid Profiles from high S1P d18:1, High AEA towards High Long Chain-Ceramides

Key candidate lipids were selected based on the CanDisc structure, and case profiles of the three top lipids were further compared. Paired analyses of the individual’s Cer22:0 and S1Pd18:1 levels show that MD/BD patients “always” had higher Cer22:0 than S1Pd18:1, except for one patient (Figure 3A). This pattern appears to be inverted in ADHD, except for two patients. One was a 37-year-old male suffering from obesity (BMI 37), diabetes mellitus, and hypertension. The other was a 26-year-old male without metabolic reason for high Cer22:0, but a serious trauma in his medical history. The Cer22:0/S1Pd18:1 ratio (Log2 difference) differed significantly between groups (Figure 3B). The case profiles of S1Pd18:1, Cer22:0, and AEA were further compared and sorted according to the S1Pd18:0 level. Despite some overlap with the healthy controls, the profile changes according to the diagnosis suggested a disease-specific pattern.

## 4. Discussion

The present pilot study shows that ADHD patients have elevated levels of plasma anandamide (AEA) and sphingosine-1-phosphate (S1Pd18:1). In contrast, MD/BD patients have increased ceramides, the latter in agreement with our previous data from an independent cohort of MD/BD patients [29]. Interestingly, ADHD patients with a comorbidity of MD/BD show elevations of both lipid groups, however to a smaller extent. S1Pd18:1 and ceramides are increased in these comorbid patients, but the individual lipids are not as much increased as in the respective mono-diagnosis group.

As some patients report beneficial effects of cannabis [23], the finding of increased plasma endocannabinoid levels appears to be surprising. Reportedly, five ADHD patients (4 ADHD+, 1 ADHD) used cannabis occasionally, but exogenous cannabis rather reduces plasma endocannabinoids [49] and does not explain the observed high plasma AEA levels. Increases in endocannabinoids in plasma are currently interpreted as an attempt to compensate for a central deficiency [20]. Indeed, brain endocannabinoids were substantially reduced in mice with an “ADHD-like” posttraumatic stress phenotype in association with hyperactivity and loss of attention [50], suggesting that central endocannabinoid deficiency may have a role in ADHD-like phenotypes [51]. High plasma levels of AEA in ADHD patients do not contradict a hypothesis of a relative endocannabinoid deficiency in these patients. In line with the notion that endocannabinoids are implicated in ADHD, genetic polymorphism of fatty acid amide hydrolase (*FAAH*; rs2295633) has been associated with ADHD [11]. FAAH is needed for AEA reuptake and metabolism, and FAAH inhibitors are suggested to increase synaptic AEA leading to beneficial therapeutic effects in models of chronic pain, anxiety, or depression [52,53,54,55,56]. The *FAAH* rs2295633 polymorphism also has been associated with posttraumatic stress disorder [57]. Other *FAAH* variants were genetically linked with addiction, anxiety, eating disorders, depression, and obesity [58,59,60,61,62,63], emphasizing the endocannabinoid system’s importance for mental health but not specifically for ADHD. Presently, it is not known whether the ADHD-associated *FAAH* variant reduces or increases reuptake or degradation of AEA. Both, loss and gain of function of *FAAH* variants have been discussed [59,64,65]. It is intriguing to speculate that central endocannabinoid deficiency might delineate a subgroup of ADHD patients who could benefit from treatment with CB1 receptor agonists. Further studies addressing this topic are needed to clarify whether endocannabinoid disturbances contribute to underlying mechanisms.

The finding of increased ceramides in MD/BP replicates the results of our previous observational study in an independent cohort of MD/BP patients [29]. It extends this by demonstrating that Cer22:0 is the most robust individual ceramide species associated with an affective disorder. Regarding AHDH, one previous study reported reduced serum sphingomyelins and long chain ceramides (Cer24:0, Cer24:1) in ADHD children, suggesting that low serum sphingomyelins reflected alterations of brain maturation in childhood ADHD [66]. Serum S1P d18:1 was also reduced in these ADHD children [66]. S1Pd18:1 has been recognized as a key signaling molecule in neuro-immunologic and other inflammatory diseases owing to the therapeutic efficacy of the S1P receptor ligand, fingolimod that works as a functional non-specific S1P receptor antagonist. Fingolimod prevented social deficits in a rat model of autism [67], and higher serum levels of S1P d18:1 were reported in rodent models after induction of anxiety-like behavior [26]. Correspondingly, S1Pd18:1 treatment appeared to increase anxiety-like behavior [26]. Owing to the limitations of rodent models of ADHD [68], it is presently unknown if fingolimod or newer S1P-receptor targeting drugs could reduce behavioral correlates of ADHD. The observed elevation of plasma S1P in our patients might imply that functional antagonists might be useful therapeutically for ADHD if the plasma levels indicated increased S1P signaling, which is presently unknown. In previous studies, S1P d18:1 plasma levels in patients with an affective disorder or neurologic diseases were not affected by the disease [29,39,69], but low S1Pd18:1 in association with high ceramides were observed in patients with dementia [70]. It has to be considered that the ages of dementia patients (avg. age 70) were higher than in the controls (avg. age 38) [70], so that the observed lipid patterns may have been influenced by age owing to the linear increase of ceramides with age [29] (Appendix A).

Differences in ages might also have affected the continuous shift of the S1P d18:1/Cer22 profiles in patients without or with MD/BP comorbidities because ADHD patients without comorbidities were about 10 years younger (Table 1). The controls covered the whole range of ages, but age differences may have affected the separation of the patient groups. Nevertheless, it is remarkable that ADHD patients with comorbidity shared features of the lipid profiles of the ADHD-only and the MD/BP-only group, suggesting that an increase of Cer22:0 (and other ceramide species) might recognize a comorbid course and that lipid profiles indeed map to psychopathological domains.

Due to the heterogeneity of drug treatments relative to the low sample size, discriminant analyses did not reveal associations with drug treatments. The change of the lipid profiles from ADHD-only to comorbid-ADHD could not be assigned to a change of drugs or additional drugs. However, the comparisons of drug effects were limited because of the diversity of the patients, diversity of drugs and drug combinations, and the relatively small sample size. Further limitations are the confounding factors of age, metabolic status, and sex. For ceramides, very high BMI (>35) is associated with high ceramides, which may explain why one of the ADHD-only patients did not fit the ADHD scheme of “high S1Pd18:1/low Cer22:0”. The other patient who had a “non-ADHD-only” lipid profile had had an accident leading to amputation of both legs and chronic pain. In addition, these two patients were taking methylphenidate (MPH) for more than 3 years. In contrast, the other patients of the ADHD-only group were newly diagnosed and had just started MPH or lisdexamfetamine. In the “MD/BD-only group,” only one of the patients did not show high long-chain ceramides. In contrast to the other MD/BP patients, this patient did not receive any drugs and had a normal body weight.

## 5. Conclusions

The present study suggests that ADHD patients have increased anandamide and S1P d18:1 plasma levels, whereas MD/BP patients display increased ceramides. Plasma S1P d18:1, Cer22:0 thus might be useful for differential diagnosis or to define subgroups of the disease, pending replication of our findings in larger samples.

## Figures and Tables

**Figure 1 biomedicines-09-01173-f001:**
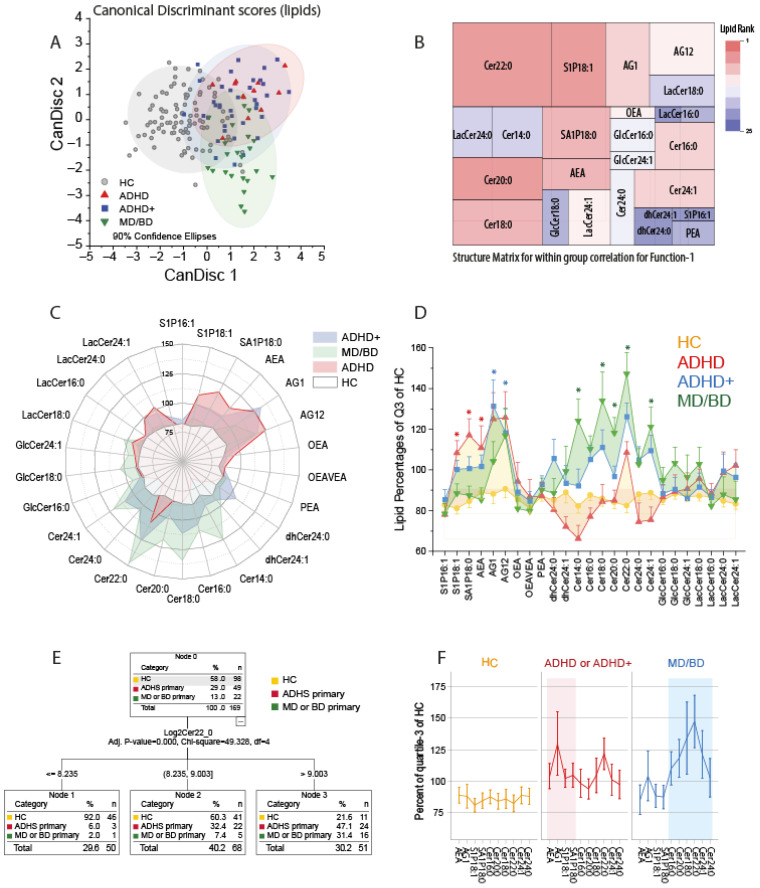
Canonical Discriminant Analysis, CHAID analysis and lipid profiles. (**A**): The scatter plots show the CanDisc scores for the first two functions in patients with attention-deficit/hyperactivity disorder (ADHD), ADHD with comorbidity of major depression (MD) or bipolar disorder (BD) (ADHD+), patients with MD/BD without ADHD, or healthy controls (HC). Each scatter shows one subject. The ellipses show the 90% confidence intervals (CI). Samples sizes: n = 12 ADHD, n = 37 ADHD+, n = 22 MD/BD, n = 98 HC. (**B**): Tree map of the CanDisc structure matrix for function-1, which mainly discriminates between patients and healthy controls. The CanDisc structure of function-2 is in Appendix A. The sizes of the rectangles and the colors reveal the relative importance of the respective lipid for the discrimination of the groups. The sizes are the within-group correlations, and the colors show the ranking. (**C**): Polar plot of the group means of lipid concentrations normalized to percentages of the 75% quantile of healthy controls. Bulges of the polar stars towards a lipid or group of lipids show that those lipids increased relative to the HC. (**D**) The area plot of the lipid profiles shows mean ± SD of normalized lipids (normalized as in (**C**)). The asterisks indicate significant differences versus HC of the respective lipid according to one-way ANOVA and posthoc *t*-test using an alpha adjustment according to Šidák, * adjusted *p* < 0.05. (**E**): Decision tree of CHAID (Chi-square automatic interaction detection) analysis. (**F**): The line graphs show the lipid profiles of the key lipids, which were selected according to the CanDisc structure matrix. ADHD and ADHD+ patients were combined.

**Figure 2 biomedicines-09-01173-f002:**
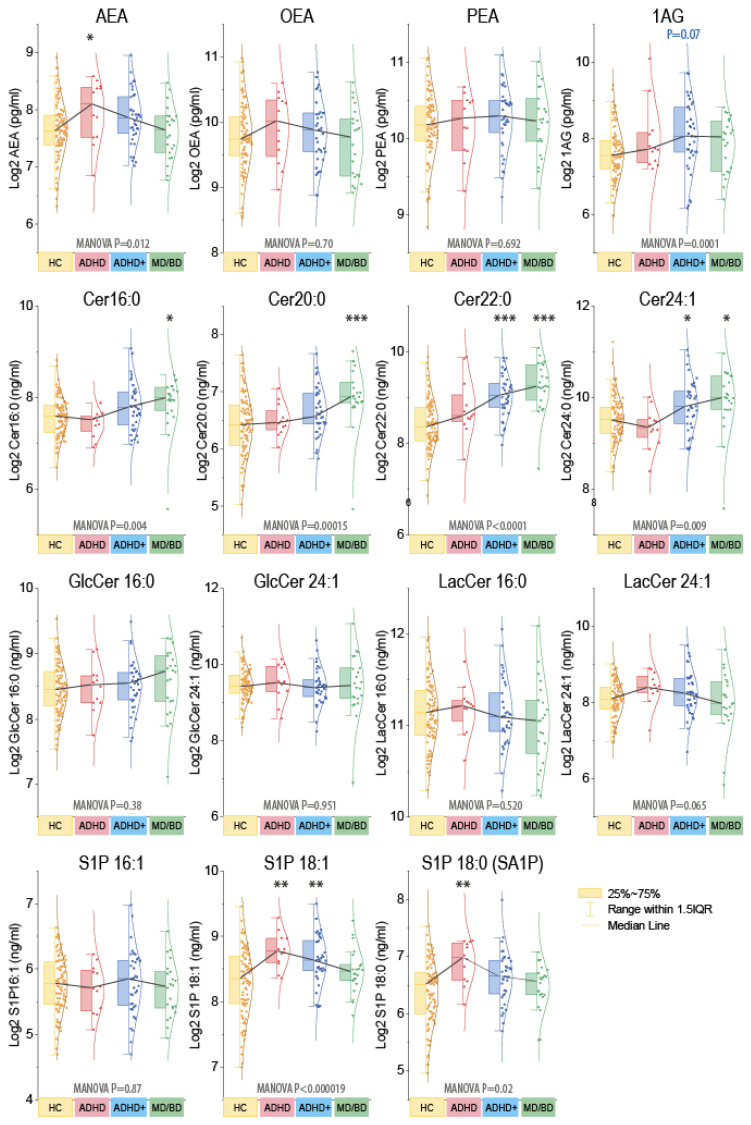
Grouped plasma concentrations of endocannabinoids and ceramides. The box plots show the interquartile range (IQR). The black line connects the medians. The scatters are the subjects, and the whiskers show the range within 1.5× the IQR. The Gauss curves show the distribution. The data are log2 transformed because lipids were log-normally distributed. Endocannabinoids are pg/mL, sphingoid bases, and ceramides ng/mL. Because of the impact of sex for endocannabinoids (Appendix A), data were compared with MANOVA for the factors “group” X “sex” and subsequent posthoc for the group using an adjustment of alpha according to Dunnett, i.e., versus HC. Asterisks indicate significant differences versus HC, * *p* <0.05, ** *p* <0.01, *** *p* <0.001. Samples sizes: n = 12 attention-deficit/hyperactivity disorder (ADHD), n = 37 ADHD with comorbidity (ADHD+), n = 22 major depression of bipolar disorder (MD/BD), n = 98 healthy controls (HC).

**Figure 3 biomedicines-09-01173-f003:**
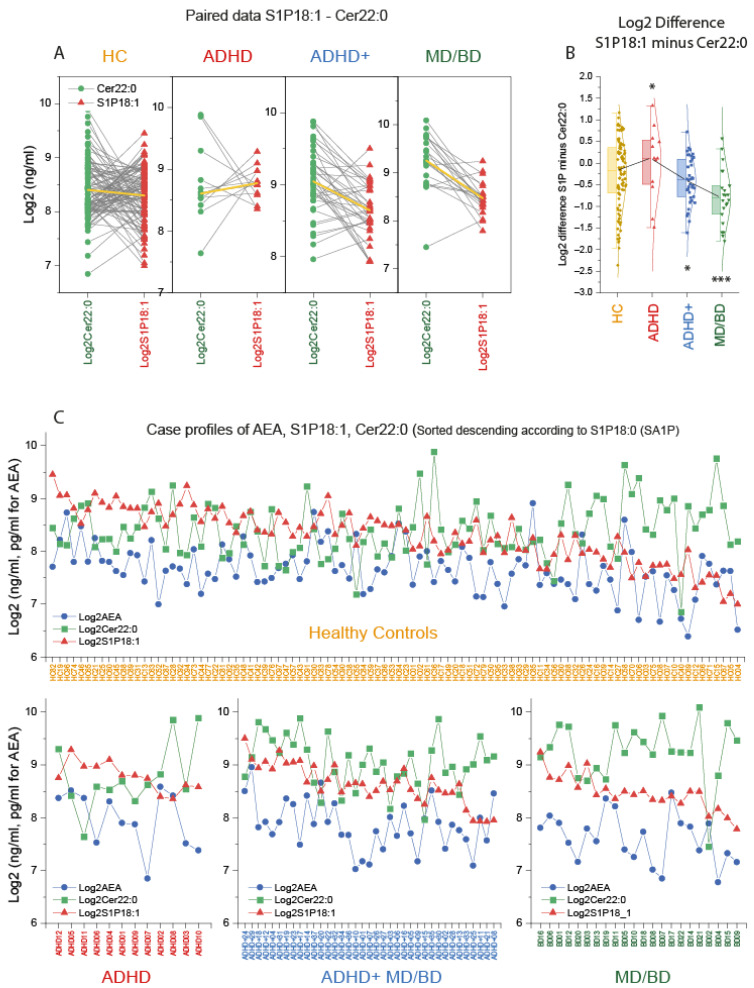
Case profiles of three key lipids: Cer d18:1/22:0, AEA, and S1P d18:1. (**A**): Paired analysis of the subjects’ log2 plasma concentration of ceramide 22:0 (Cer22:0) and sphingosine-1-phosphate (S1P d18:1) in patients with attention-deficit/hyperactivity disorder (ADHD), ADHD with comorbidity of major depression (MD) or bipolar disorder (BD) (ADHD+), patients with MD/BD without ADHD, or healthy controls (HC). The yellow lines show the group means. (**B**): The box plots show the log2 difference of S1P d18:1 and Cer22:0 concentrations per subject (S1P d18:1 minus Cer22:0). The box is the interquartile range (IQR), the black line connects the medians. The scatters are the subjects, and the whiskers show the range within 1.5× the IQR. The Gauss curves show the distribution. According to Dunnett, data were compared with one-way ANOVA and subsequent posthoc analysis versus HC with adjustment of alpha. * *p* <0.05, *** *p* <0.001. (**C**): Log2 concentrations for each subject of the candidate lipids, anandamide (AEA), ceramide 22:0 (Cer22:0) and sphingosine-1-phosphate (S1P d18:1). The cases are sorted from left (high) to right (low) according to the S1P d18:0 (sphinganine-1-phosphate) level, which is not depicted because the information is redundant to S1P d18:1. There are two ADHD patients, who according to the lipid profile, would better fit into the ADHD+ group.

**Table 1 biomedicines-09-01173-t001:** Demographic data of healthy controls (HC) and patients with ADHD, ADHD+. Or MD/BD.

		Female				Male			
		HC	ADHD	ADHD+	MD/BD	HC	ADHD	ADHD+	MD/BD
Mean age (years)		35.45	26.00	36.67	43.36	36.88	29.00	36.23	47.00
Age range (years)		21–65	20–32	20–66	26–67	23–66	26–34	26–54	26–59
Age class	20–30	27	5	6	2	12	3	6	2
	31–40	20	2	4	3	12	2	10	1
	41–50	8	0	1	2	3	0	3	1
	>50	10	0	4	4	6	0	3	7
Mean BMI		22.82	25.39	23.68	24.58	23.61	25.82	27.50	29.43
BMI range		18.8–36.1	20.0–35.7	19.2–35.9	16.4–35.5	18.8–28.1	19.8–37.0	19.7–38.8	22.2–46.4
BMI class	≤21.0	24	3	6	3	5	1	1	0
	21.1–24.0	23	1	4	3	14	2	5	1
	24.1–28.0	13	1	3	3	13	0	7	5
	28.1+	5	2	2	2	1	2	9	5

Abbreviations: HC, healthy control; ADHD, attention deficit hyperactivity disorder; MD, major depression, BD, bipolar disorder ADHD+, ADHD with comorbidity of MD/BD.

**Table 2 biomedicines-09-01173-t002:** Comorbidities of healthy controls (HC) and patients with ADHD, ADHD+ (ADHD plus MD/BD or MD/BD.

Sex and Comorbidities		HC	ADHD	ADHD+	MD/BD
Sex	male	33	5	22	11
	female	65	7	15	11
	total	98	12	37	22
Smoking	yes	6	4	8	10
	no	92	8	29	12
Type-II diabetes mellitus	yes	0	1	2	4
	no	98	11	35	18
Hypertension	yes	1	1	7	6
	no	97	11	30	16
L-Thyroxin supplementation	yes	0	1	11	2
	no	98	11	26	20

**Table 3 biomedicines-09-01173-t003:** Electroconvulsion therapy (ECT) and drug treatments.

ECT and Drug Treatment		HC	ADHD	ADHD+	MD/BD
Electroconvulsion therapy	yes	0	0	1	1
	no	98	12	36	21
ADHD (MPH, Lisdexamfetamine, Atomoxetine, Guanfacine)	HC	98	0	0	0
	yes	0	7	16	0
	no	0	5	21	22
SNRI, NDRI (Venlafaxine, Bupropion)	HC	98	0	0	0
	yes	0	0	0	10
	no	0	12	37	12
TCA & related compounds	HC	98	0	0	0
	yes	0	0	3	4
	no	0	12	34	18
SSRI (Sertraline, Citalopram, Paroxetine)	HC	98	0	0	0
	yes	0	0	1	3
	no	0	12	36	19
Atypical antipsychotics (Quetiapine, Risperidone, Aripiprazole)	HC	98	0	0	0
	yes	0	0	3	7
	no	0	12	34	15
Antiepileptic (Lamotrigine, Valproic acid)	HC	98	0	0	0
	yes	0	0	0	2
	no	0	12	37	20

Abbreviations: MPH, methylphenidate; SNRI, selective serotonin and norepinephrine reuptake inhibitor; NDRI, norepinephrine and dopamine reuptake inhibitor; TCA, tricyclic antidepressant; SSRI, selective serotonin reuptake inhibitor.

## Data Availability

Data generated within this study are presented in the paper or supplement.

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
