# Peer review of "Sphingolipid and Endocannabinoid Profiles in Adult Attention Deficit Hyperactivity Disorder"

_biomedicines, 2021, doi:10.3390/biomedicines9091173_

Round 1
Reviewer 1 Report
Because ADHD is prone to secondary disorders such as depression, bipolar disorder, and anxiety disorders, it is necessary to conduct a comprehensive evaluation and diagnosis through various tests and interviews. The symptoms of these mental disorders are very similar, and care must be taken when administering medication, and indicators to identify these symptoms are extremely important. The present study is highly commendable because it clarified the characteristics of ADHD and its complications using blood lipids as an indicator by classifying various factors such as gender, smoking, and medication use from data of about two hundred people including healthy controls excluding children. In recent years, it has been suggested that the high concentration of free fatty acids in the blood lipids of children with autism spectrum disorder is also associated with indicators of "social impairment in children with autism spectrum disorder," and this study has a high affinity with such research and is extremely important in the study of psychiatric disorders and abnormal lipid metabolism. The protocol of the study was clearly stated and the data analysis was carefully conducted, and the results obtained are highly reliable. If the causes of the differences in blood lipid distribution identified in this study can be identified, or if the mechanism of the function of the resulting free fatty acids can be elucidated, it is hoped that new therapeutic agents and nutritional approaches to ADHD targeting them can be established. Therefore, the reviewer believes that this research is appropriate for adoption. How about a few suggestions for correction?
1. The quality of the figures is a little poor, so please replace them with better ones.In particular, the letters and numbers in Figure 1 are difficult to read.
2. The plots are not normally distributed due to the small sample size, but the reviewer feels uncomfortable with the use of normal distribution.
The amount of lipids in the blood may be affected by the previous meal, but the reviewer felt there was not enough information on this point. He would like to see a description of what restrictions were or were not placed on the person taking the measurements.
Author Response
- The quality of the figures is a little poor, so please replace them with better ones.In particular, the letters and numbers in Figure 1 are difficult to read.
Thank you for the positive evaluation of our manuscript. We have increased the size of the figures as far as possible with the Journal's template. High resolutions of the figures are included in the supplementary zip file. We have increased the fonts in figure 1.
- The plots are not normally distributed due to the small sample size, but the reviewer feels uncomfortable with the use of normal distribution.
The amount of lipids in the blood may be affected by the previous meal, but the reviewer felt there was not enough information on this point. He would like to see a description of what restrictions were or were not placed on the person taking the measurements.
The inclusion and exclusion criteria are given in the Methods paragraph. There were no other exclusion criteria or restrictions such as in terms of eating or drinking.
Reviewer 2 Report
The authors present a manuscript work on the lipid and endocannabinoid profile of some mental disorders. The work is well designed experimentally and obtains interesting conclusions. The manuscript is appropriate for publication, although some minor aspects must be corrected or explained previously.
The sample is heterogeneous for unrelated diseases, and sometimes the n for each disease is very small.
Table 1a should indicate the dispersion for age (not just the mean).
In figure 1 the letters are too small in many cases to be read correctly. The figure must be enlarged, or divided in two figures.
The authors cite several of their works, but it would be sufficient to cite only some of them (they are overlapping) (https://doi.org/10.1038/s41598-020-71879 x ; https://doi.org/10.1186/s40478-017-0446-4; https://doi.org/10.1016/j.metabol.2019.04.002). Furthermore, in many cases the methodological part is identical (literal), so citing or description could be avoided. In other cases, literal matches are found with works that are not cited (doi: 10.1038/s41598-019-38865-4; https://doi.org/10.3390/jcm8070971; doi:10.3390/ijms19051390; doi:10.3390/cancers12071753; doi:10.3390/nu12113248) .
In the discussion, some therapy based on the results could be considered as perspective.
Author Response
The authors present a manuscript work on the lipid and endocannabinoid profile of some mental disorders. The work is well designed experimentally and obtains interesting conclusions. The manuscript is appropriate for publication, although some minor aspects must be corrected or explained previously.
The sample is heterogeneous for unrelated diseases, and sometimes the n for each disease is very small.
Table 1a should indicate the dispersion for age (not just the mean).
Thank you for evaluation of our manuscript and your suggestions.
As requested, we have added in Table 1 the range of ages and range of BMI (highlighted in red letters). In addition, we have included a column of the ages and BMI in Supplementary Table S2 in addition to the age class and BMI class.
In figure 1 the letters are too small in many cases to be read correctly. The figure must be enlarged, or divided in two figures.
We have increased the size of the figures as far as possible with the Journal's template. We have increased the fonts in figure 1. High resolution tif images of the figures are included in the supplementary zip file.
The authors cite several of their works, but it would be sufficient to cite only some of them (they are overlapping) (https://doi.org/10.1038/s41598-020-71879 x ; https://doi.org/10.1186/s40478-017-0446-4; https://doi.org/10.1016/j.metabol.2019.04.002). Furthermore, in many cases the methodological part is identical (literal), so citing or description could be avoided. In other cases, literal matches are found with works that are not cited (doi: 10.1038/s41598-019-38865-4; https://doi.org/10.3390/jcm8070971; doi:10.3390/ijms19051390; doi:10.3390/cancers12071753; doi:10.3390/nu12113248) .
As suggested, we have reduced the number of references. The description of the method of lipid analysis is similar to our previous studies, in which similar analyses were done. We would like to keep a short description in the manuscript although it is similar, because if there were only a reference a reader has to search further manuscripts to find the method.
In the discussion, some therapy based on the results could be considered as perspective.
The sentences addresses putative therapeutic implications of cannabinoids is now highlighted in red letters. A sentence addressing putative targeting of S1P is added by WORD's track changes. We find it too speculative to deduce further putative therapeutic options on the bases of the observed lipid plasma profiles in the present study.